# Elementary Open Quantum States

**Janos Polonyi** *,† and **Ines Rachid** †

Physics and Engineering Department, Strasbourg University, CNRS-IPHC, 23 Rue Du Loess, BP28, CEDEX 2, 67037 Strasbourg, France; ines.rachid@iphc.cnrs.fr
* Correspondence: polonyi@iphc.cnrs.fr
† These authors contributed equally to this work.

**Abstract:** It is shown that the mixed states of a closed dynamics supports a reduplicated symmetry, which is reduced back to the subgroup of the original symmetry group when the dynamics is open. The elementary components of the open dynamics are defined as operators of the Liouville space in the irreducible representations of the symmetry of the open system. These are tensor operators in the case of rotational symmetry. The case of translation symmetry is discussed in more detail for harmonic systems.

**Keywords:** open systems; elementary particle; irreducible representation

**PACS:** 03.65.Fd

## 1. Introduction

One of the surprises of quantum mechanics is the emergence of elementary excitations, the discrete building blocks of the state of a dynamical system. This results from representing the physical states by vectors in a Hilbert space [1]. In fact, the symmetries of the dynamics are realized in the Hilbert space in such a manner that they can be broken up into the direct sum of irreducible representations. The importance of the irreducibility is that the subspaces of these representations are the smallest linear subspaces, which remain closed during the time evolution. Hence, the vectors of the irreducible representations define the elementary excitations.

One would expect that this structure holds only for closed systems. The elementary bare excitations of relativistic quantum field theory are dressed by vacuum polarization and are subjects of an open dynamics. An electron, for instance, polarizes a virtual particle–antiparticle cloud from the vacuum according to the (bare) perturbation expansion. This phenomenon is not limited to relativistic dynamics, as one can easily see in the case of a polaron, a charge inserted into a polarizable medium. These examples suggest that an elementary excitation of a closed system dresses and becomes a particular combination of closed elementary excitations when its interaction with its environment is taken into account.

Furthermore, one expects that an open dynamics defines its own elementary states, referring to phenomena within the scale window where the effective open dynamics is valid. For instance, the vacuum polarization cloud of an electron follows the bare electron without any change in its internal structure as long as the electron is subjected to weak enough external forces.

The present work is an attempt to see to what extent these expectations are fulfilled. One notices at the very beginning that the elementary states of an open system are a more involved mathematical concept than that of a closed dynamics. This has actually nothing to do with the openness of the dynamics. The mixed states of an open system are represented by density matrices, which can be constructed along two physically different routes. On the one hand, an uncertainty about the pure state, expressed by a particular probability distribution over several pure states, leads to the density matrix representation. On the

other hand, mixed states are generated when the system interacts with its environment. The first view of the mixed states shows clearly that they are not elementary. However, a particular dynamics, appearing in the second view, may generate its own irreducible representation spaces.

Some difficulties appear in the definition of the elementary open states because the density matrices form a rather complicated subset of the Liouville space, consisting of Hermitian, unit trace, positive operators. A possible formal way of avoiding the problem is to restrict the scope of the discussion to elementary open state components, defined as irreducible subspaces of the Liouville space. The obvious disadvantage of this strategy is that not all irreducible spaces represent possible physical states.

This problem can partially be removed by noting that any symmetry group $G$ of a closed dynamics, $|\psi\rangle \to U(g)|\psi\rangle$ $g \in G$, extends to the symmetry $G \otimes G$ acting on the Liouville space, $A \to \mathcal{U}(g_-, g_+)A = U(g_-)AU^\dagger(g_+)$ with $g_\pm \in G$, since it can be realized on the bras and the kets independently. Such a doubled symmetry is reduced back to at most the diagonal subgroup $g_+ = g_- \in G$ when the dynamics becomes open. Such a symmetry breaking by the open interaction channels can be used to establish the relation between the closed and open irreducible subspaces. The reduplication of the symmetry has already been spotted in a number of open systems, in particular at finite temperature [2], in hydrodynamics [3,4], in the extension of the BRST symmetry [5], and in identifying Goldstone bosons [6–8]. The present work aims at a simpler problem, the use of the reduplicated symmetry for the construction of elementary states in nonrelativistic (first quantized) quantum mechanics.

The irreducible spaces of $G \otimes G$ in the Liouville space define the elementary mixed components. These can be split into the direct sum of the irreducible spaces of the diagonal subgroup $G$, the symmetry group of the open dynamics. These smaller spaces contain the elementary open components. This structure is exploited in this work by considering a closed elementary state within an irreducible space of $G \otimes G$ and following its time evolution during the open dynamics. The importance of the open elementary state components manifests itself by noticing that the irreducible space of the diagonal subgroup $G$ remains closed during the time evolution. Hence, the decomposition of the mixed states into the sum of open elementary components offers a representation in terms of elementary blocks, evolving independently of each other.

This structure is introduced below, starting with a brief introduction of the symmetry group $G \otimes G$ of the mixed state of a closed dynamics in Section 2, followed by Section 3, with the reduction of the symmetry to a diagonal subgroup by the open interaction channels. The irreducible spaces, containing the elementary state components, are introduced in Section 4. This construction is outlined briefly for rotational and translational symmetry in Section 4. The time dependence of the elementary open components of the latter case is presented in Section 5. The summary of the results is given in Section 6.

## 2. Closed Systems

We start with a closed quantum system whose states are represented by the vectors of a Hilbert space $\mathcal{H}$ and dynamics is given by the Schrödinger equation, $i\hbar\partial_t|\psi\rangle = H|\psi\rangle$. The ket $|\psi\rangle \in \mathcal{H}$ and bra $\langle\psi| \in \mathcal{H}^\dagger$ stand for the same physical state. The sum over the basis vectors in the expectation value in a pure state,

$$\langle\psi|A|\psi\rangle = \sum_{m,n}\langle\psi|m\rangle\langle m|A|n\rangle\langle n|\psi\rangle, \tag{1}$$

represents the sum over quantum fluctuations, filtered by a given basis set. Since the coefficient of $\langle m|A|n\rangle$ in the sum, the pure state density matrix, is factorized, $\langle n|\rho|m\rangle = \langle n|\psi\rangle\langle\psi|m\rangle$, the quantum fluctuations of the bra and the ket components are independent of each other in a pure state. The bra and the ket are related by Hermitian conjugation; hence, it is sufficient to follow the time dependence only for one of them.

We further suppose that a symmetry group $G$ of the closed dynamics is represented by the basis transformations of $\mathcal{H}$; there are unitary operators acting on the pure states, $|\psi\rangle \to U(g)|\psi\rangle$, $\langle\psi| \to \langle\psi|U^\dagger(g)$, and operators $A \to U(g)AU^\dagger(g)$, $g \in G$ in such a manner that they preserve the scalar product, the matrix elements, and leave the form of the Schrödinger equation invariant, $H = U(g)HU^\dagger(g)$. Thus, the symmetry group of the closed dynamics on pure states is $G$.

The mixed states are represented by density matrices, represented by certain elements of the Liouville space, $\mathcal{A} = \mathcal{H} \otimes \mathcal{H}^\dagger$, equipped by the scalar product $\langle\langle A|A'\rangle\rangle = \text{Tr}[A^\dagger A']$. The density matrices belong to $\mathcal{A}_\rho$, a convex subset of $\mathcal{A}$, containing the positive, Hermitian operators with unit trace. The elements of $\mathcal{A}$ are called state components. The expectation value of the observable $A = A^\dagger$ in the state $\rho \in \mathcal{A}_\rho$ is given by the Liouville space scalar product, $\text{Tr}[\rho A^\dagger] = \langle\langle A|\rho\rangle\rangle$.

In the case of a symmetry, we introduce the extended basis transformations, which act with different symmetry group elements on the bra and the ket, $|\psi\rangle \to U(g_+)|\psi\rangle$, $\langle\psi| \to \langle\psi'|U^\dagger(g_-)$ and transform the operators as $A \to \mathcal{U}(g_-, g_+)A = U(g_-)AU^\dagger(g_+)$. It is easy to see that the scalar product is preserved by these transformations and that the Neumann equation, $i\hbar\partial_t\rho = [H, \rho]$, remains invariant as well,

$$U(g_-)[H, \rho]U^\dagger(g_+) = [H, U(g_-)\rho U^\dagger(g_+)], \tag{2}$$

reflecting the independence of the bra and ket dynamics of closed dynamics. Hence, the symmetry group of the mixed state descriptions of a closed dynamics is $G \otimes G$.

## 3. Open Systems

We extend our description from the observed system to its environment by assuming that the full system obeys a closed dynamics, defined by the total Hamiltonian, $H_{tot} = H_s + H_e + H_i$, where the terms in the sum correspond to the system, the environment, and their interactions. The pure states make up the direct product Hilbert space, $\mathcal{H}_{tot} = \mathcal{H}_s \otimes \mathcal{H}_e$, where the first and the second factor denote the space of system and environment states, respectively. The state of the observed system is given by the reduced density matrix, $\rho_s(t) = \text{Tr}_e[\rho(t)]$, where $\rho(t) = e^{-\frac{i}{\hbar}H(t-t_i)}\rho(t_i)e^{\frac{i}{\hbar}H(t-t_i)}$. A further assumption is that there are no correlations between the system and the environment in the initial state, i.e., the initial density matrix is factorized, $\rho(t_i) = \rho_{s,i} \otimes \rho_{e,i}$.

Let us consider a symmetry group $G$ of the full closed dynamics, represented by basis transformations, $H_{tot} = U(g)H_{tot}U^\dagger(g)$. While the Neumann equation of the full dynamics is symmetrical with respect to the extended transformations,

$$U(g_-)\rho(t)U^\dagger(g_+) = e^{-\frac{i}{\hbar}H(t-t_i)}U(g_-)\rho(t_i)U^\dagger(g_+)e^{\frac{i}{\hbar}H(t-t_i)}, \tag{3}$$

the symmetry group of the effective system dynamics is usually more restricted. To find out the latter, we assume that $U(g)$ acts without mixing the system and the environment states, $U(g) = U_s(g) \otimes U_e(g)$. The invariance of the trace under basis transformation implies the identity $\rho_s = \text{Tr}_e[\rho] = \text{Tr}_e[U_e\rho U_e^\dagger]$, holding for any unitary environment operator, $U_e$. From this identity follows:

$$U_{s,-}\rho_s(t)U_{s,+}^\dagger = \text{Tr}_e[e^{-\frac{i}{\hbar}H(t-t_i)}U_{s,-}\rho_{s,i}U_{s,+}^\dagger \otimes U_e\rho_{e,i}U_e^\dagger e^{\frac{i}{\hbar}H(t-t_i)}] \tag{4}$$

if $U_{s,\pm}$ and $U_e$ are symmetries of each term in the Hamiltonian one-by-one, $H_s = U_{s,\pm}H_sU_{s,\pm}^\dagger$, $H_e = U_eH_eU_e^\dagger$, and $H_i = U_{s,\pm} \otimes U_eH_iU_{s,\pm}^\dagger \otimes U_e^\dagger$. The necessity of the last condition shows that the symmetry with respect to the extended transformation, $\rho_s \to U_{s,-}\rho_sU_{s,+}^\dagger$ with $U_{s,+} \neq U_{s,-}$, is usually no longer a symmetry in the presence of system–environment interactions. The condition of preserving the full symmetry group for the diagonal basis transformations, $g_+ = g_-$, is the symmetry of the environment initial condition, $\rho_{e,i} = U_e(g)\rho_{e,i}U_e^\dagger(g)$.

The system–environment entanglement induces a nonfactorizable density matrix; hence, a signature of the openness of the dynamics is that the quantum fluctuations of the bra and the ket components are correlated. Such correlations reduce the symmetry of the mixed state description to the diagonal subgroup of $G \otimes G$, $g_+ = g_-$. Therefore, such a symmetry breaking serves as an indication of an open dynamics.

### 4. Elementary State Components

The elementary pure states are defined as vectors of the irreducible representations of $G$ in $\mathcal{H}$ because these representations provide the smallest linear subspaces, which are kept closed by the time dependence. This concept plays a double role in quantum mechanics. On the one hand, the kinematical use is the construction of the possible physical systems by combining the elementary degrees of freedom. On the other hand, by assuming that the group $G$ is a symmetry of the dynamics, the time dependence can substantially be simplified by taking into account the closeness of the representation spaces during the time evolution.

The definition of the elementary pure state can formally be generalized for the mixed states of closed and open dynamics: the elementary closed (open) mixed state components are the elements of the irreducible representation of $G \otimes G$ ($G$) in the Liouville space. While these components remain in a restricted linear space during the time evolution as in the case of pure elementary states, their kinematical role is more restricted. In fact, the physical density matrices form a convex hull rather than a linear subspace in the Liouville space owing to their positivity. Thus, we cannot freely form the linearly superposition of the elementary components since the positivity, a global condition, must be observed.

We have two different ways to split the Liouville space into the direct sum of the irreducible representations by using the symmetry of either the closed or the open dynamics. Since $G \subset G \otimes G$, the representation spaces of the closed symmetry contain several representation spaces of the open dynamics. One can introduce two classes of elementary open components, the operators with a nonvanishing (vanishing) trace making up the first (second) class. Each physical state contains at least one first class component, which preserves its norm, and the second class components might be generated during the time evolution without any constraint on their weight in the state. Special attention is needed for state components in infinite three-space with an ill-defined trace, such as plane waves. The master equation of the Lindblad structure preserves the total probability only if constructed by means of bounded operators. The unbounded coordinate operator may generate a time-dependent weight for the components, which can be understood as the result of a nonvanishing particle flux at $|x| \to \infty$ [9].

We briefly discuss the cases of rotations, $G = SO(3)$, and translations, $G = R^3$, below.

*4.1. Rotations*

The elementary pure state vectors are in the irreducible rotational multiplets, $\mathcal{H}_\ell$, of dimension $2\ell + 1$. The elementary mixed components belong to the linear subspaces $\mathcal{A}_{n_-,\ell_-,n_+,\ell_+}$, spanned by the dyadic products $|n_-,\ell_-,m_-\rangle\langle n_+,\ell_+,m_+|$, where $n$ stands for the rotational independent quantum numbers. The elementary state components of the rotationally invariant open dynamics are given by the subspaces, $\mathcal{A}_{n,\ell}$, spanned by the tensor operators $\{T_m^{(n,\ell)}\}$. The structure of an elementary open component in terms of the elementary components of the closed dynamics,

$$
\begin{aligned}
&\langle\langle T_m^{(n,\ell)} || n_-,\ell_-,m_-\rangle\langle n_+,\ell_+,m_+|\rangle\rangle \\
&= \langle n_+,\ell_+,m_+| T_m^{(\ell)\dagger} |n_-,\ell_-,m_-\rangle \\
&= \langle n_-,\ell_-,m_-| T_m^{(n,\ell)} |n_+,\ell_+,m_+\rangle^* \\
&= (\ell_+,\ell,m_+,m|\ell_-,m_-)\langle\langle n_-,\ell_-| T^{(n,\ell)} |n_+,\ell_+\rangle\rangle^*
\end{aligned}
\tag{5}
$$

is given by the Wigner–Eckart theorem, $\langle\langle n_-,\ell_-| T^{(n,\ell)} |n_+,\ell_+\rangle\rangle$ being the reduced matrix element. The structure of an elementary open component in terms of the closed irreducible

subspaces is controlled by the reduced matrix element. The more detailed structure within the irreducible spaces is given by the Clebsch–Gordan coefficient. The importance of the tensor operator subspaces, $\mathcal{A}_{n,\ell}$, is that they remain closed during the time evolution of a rotationally invariant open dynamics.

The basis transformation from the open to the closed elementary components describes the dressing of a state by the environment. Consider for the sake of example a polaron, a point charge inserted into a polarizable medium and dressed by the induced polarization cloud in the medium, its environment. The density matrix of the elementary polaron states is the tensor operators, $T_m^{(n,l)}$. The decomposition of the polaron into the elementary closed mixed state components, $|n_+, l_+, m_+\rangle\langle n_-, l_-, m_-|$, is given by (5). When the point charge–medium interaction is suddenly switched off, then each elementary closed component is conserved one-by-one. The basis transformation from the closed to the open elementary components appears in the opposite thought experiment, where we start with a closed component, switch on the point charge–medium interactions, and find in the resulting mixed state component a linear sum of different elementary open components.

*4.2. Translations*

In the case of translation-invariant dynamics, the irreducible subspaces of the closed dynamics are one-dimensional. The elementary pure states, $|q\rangle$, are characterized by their wave vector $q$, and the elementary mixed state components of the closed dynamics are the dyadic products $|q_+\rangle\langle q_-|$. In the coordinate representation, we have $A(x_+, x_-) = \langle x_+|A|x_-\rangle$, and the parametrization $x = (x_+ + x_-)/2$ and $x_d = x_+ - x_-$ turns out to be useful. The elementary mixed state components of the closed dynamics are the dyadic product, $D_{k_+,k_-}(x_+, x_-) = e^{ik_+x_+ - ik_-x_-}$. The irreducible subspace of the open dynamics, $\mathcal{A}_q$, consists of $A_q(x, x_d) = e^{iqx}\chi(x_d)$ with some $\chi(x_d)$. It is natural to use a continuous quantum number $k$ to define a translation-independent basis in $\mathcal{A}_q$,

$$A_{q,k}(x, x_d) = e^{iqx + ikx_d} = D_{\frac{q}{2}+k, \frac{q}{2}-k}(x, x_d). \tag{6}$$

This equation, the change of basis between the elementary closed and open components, is the analog of (5) with $(n, \ell) \leftrightarrow (k, q)$. An elementary open component can be written in the form of a Fourier integral,

$$A_q(x, x_d) = e^{iqx} \int \frac{d^3k}{(2\pi)^3} \chi_k D_{\frac{q}{2}+k, \frac{q}{2}-k}(x, x_d), \tag{7}$$

Note that the quantum number $q$ is not related to the momentum. In fact, the momentum operator is the sum of two terms:

$$p_\pm = \frac{\hbar}{i}\nabla_\pm = \frac{\hbar}{i}\left(\frac{1}{2}\nabla \pm \nabla_d\right) \tag{8}$$

with $\nabla_\pm = \partial/\partial x_\pm$. The contribution of the first term to the expectation value of a momentum-dependent observable is vanishing for normalizable states, and a simplified $q$-independent momentum operator, $-i\hbar\nabla_d$, can be used in coordinate-independent observables. In the case of translation noninvariant observables, the term with $\nabla$ is needed, and it takes care of the coordinate dependence of the observables.

## 5. Harmonic, Translation-Invariant Open Dynamics

Let us consider the time dependence of the elementary open components of a translation-invariant open dynamics in more detail. We simplify the discussion by restricting our attention to a simple dynamics where the equation of motion for the reduced system density matrix is local in time and contains only the first time derivative. A further simplification is that we chose harmonic dynamics where the equation of motion contains quadratic

terms in the canonical operators. The most general harmonic translation-invariant master equation of the Lindblad form [10] is:

$$
\begin{aligned}
\partial_t \rho \quad = \quad & \left[ \frac{i\hbar}{m} \boldsymbol{\nabla} \boldsymbol{\nabla}_d + \frac{i}{\hbar} f x_d - \frac{d_0 + d_2 \nu^2 - 2m\nu\xi}{2\hbar} x_d^2 \right. \\
& \left. + i \left( \frac{d_2 \nu}{m} - \xi \right) x_d \boldsymbol{\nabla} - \nu x_d \boldsymbol{\nabla}_d + \frac{\hbar d_2}{2m^2} \boldsymbol{\nabla}^2 \right] \rho,
\end{aligned}
\tag{9}
$$

with $\boldsymbol{\nabla} = \partial/\partial x$ and $\boldsymbol{\nabla}_d = \partial/\partial x_d$. The first two terms correspond to the Neumann equation of a closed system under the influence of a dragging force $f$. The parameter $\nu$ generates Newton's friction force; $d_0 > 0$ and $d_2 > 0$ control the decoherence; $\xi$ is the coefficient of a total time derivative term in the effective Lagrangian and represents no dynamical process [11]. The positivity of the density matrix is preserved for weak friction as long as $m^2(\nu^2 + 4\xi^2) \leq 2d_0 d_2$. The solution of this master equation or one of its simplified forms has been studied extensively [12,13], and here, we confine ourselves only to some simple remarks about the time dependence of the elementary open components.

The elementary state components of the translation-invariant dynamics are labeled by the wave vector $q$, which might be complex. It is easy to see that the master Equation (9) preserves such an $x$-dependence; hence, the time evolution keeps the subspaces $\mathcal{A}_q$ of the form $\rho(x, x_d) = e^{iqx}\chi(x_d)$ closed. This expression might be used in the full coordinate space or only in a restricted region, depending on the actual problem. The master equation for the factor $\chi(x_d)$,

$$
\begin{aligned}
\partial_t \chi \quad = \quad & \left( i\frac{\hbar q}{m} - x_d \nu \right) \boldsymbol{\nabla}_d \chi \\
& + \left[ \frac{\hbar q^2 d_2}{2m^2} + \left( \frac{f}{\hbar} + \frac{q d_2}{m}\nu - q\xi \right) i x_d - \frac{d_0 + d_2 \nu^2 - 2m\nu\xi}{2\hbar} x_d^2 \right] \chi,
\end{aligned}
\tag{10}
$$

can easily be solved,

$$
\chi(t, x_d) = \chi_i \left( x_d e^{-\nu t} - \frac{\hbar q}{m\nu}(1 - e^{-\nu t}) \right) e^{a_q(t) + i k_q(t) x_d - \frac{d}{2}(1 - e^{-2\nu t}) x_d^2},
\tag{11}
$$

where the initial condition $\chi_i(x_d) = \chi(0, x_d)$ is used and:

$$
\begin{aligned}
a_q(t) \quad = \quad & -\frac{1}{2m\nu^2}[2iqf(\nu t - 1 + e^{-\nu t}) + \hbar q^2 \xi(1 + 2ie^{-\nu t} + e^{-2\nu t})] \\
& -\frac{\hbar q^2}{4m^2\nu^3}[d_0(-e^{-2\nu t} + 4e^{-\nu t} - 3 + 2\nu t) + d_2\nu^2(1 - e^{-2\nu t})], \\
k_q(t) \quad = \quad & \frac{1 - e^{-\nu t}}{\hbar\nu}(f - i\hbar q\xi e^{-\nu t}) - i\frac{q}{2m\nu^2}[d_0(1 - e^{-\nu t})^2 - d_2\nu^2(1 - e^{-2\nu t})], \\
d \quad = \quad & \frac{d_0 + d_2\nu^2 - 2m\nu\xi}{2\hbar\nu}.
\end{aligned}
\tag{12}
$$

We make here only a few general comments about these elementary components, and their application to simple problems was discussed in [14]. The solution gives the elementary components as the product of two factors: one is the initial condition, $\chi_i(x_0)$, considered at the initial coordinate of the characteristic passing $x_d$ at $t$. This factor displays the presence of the diffusive forces: the information of the initial state is reduced to a single number, $\chi_i(i\hbar q/m\nu)$, as $t \to \infty$. The other multiplicative factor, written in the form of an exponential function, represents the physical processes building up on the characteristic

curve during the time evolution and contains three terms. The $\mathcal{O}(x_d^0)$ piece provides a time-dependent weight factor,

$$e^{a_q} \approx e^{-\frac{\hbar q}{m}\left(ik_f + \frac{d_0}{2mv^2}q\right)t},$$ (13)

for long time, $tv \gg 1$, where the wave vector $k_f = f/\hbar v$ corresponds to the velocity $v_f = \hbar k_f/m$, which balances the dragging and the friction force, $f = mvv_f$. The $\mathcal{O}(x_d)$ part contributes by $\hbar k_q$ to the momentum, $p = \hbar(\nabla/2 + \nabla_d)/i$, and the $\mathcal{O}(x_d^2)$ term describes a Gaussian decoherence, building up gradually during the time evolution. The time scale of both the dissipation (memory loss) and the decoherence is $1/v$.

It is instructive to look into the structure of the time-dependent weight factor, which results from two processes: one is driven by the external force and the other by the environment, represented by the first and the second term in the parentheses of the exponent of (13), respectively. Let us first take the case of a real wave vector when the contribution to the probability density of the coordinate, $\rho(x,0)$, is a standing wave. The decoherence, implied by the presence of $d_0$ in the exponent of (13), tends to suppress this state component, and the concomitant smearing of the probability distribution is slowed down by increasing $v$, the strength of the dissipative force. The case of the imaginary wave number, $q = iq_i$ with real $q_i$, is more interesting because $\rho(x,0)$ can be interpreted as a probability distribution displaying an exponential coordinate dependence, $e^{-q_i x}$ and used in the coordinate space segment $x_{\pm,j}\text{sign}(q_{i,j}) \geq 0, j = 1, 2, 3$. To support such an inhomogeneous particle density against diffusion, one needs an outflux of particles at $x = 0$. Such an outflux decreases the norm of this state component because the contribution of the compensating influx to the norm at $x_{\pm,j}\text{sign}(q_{i,j}) = \infty$ is suppressed. To find the necessary flux to keep the state in equilibrium, we chose the external force to compensate this flux in (13), $qk_f = -q_i^2 d_0/2mv^2$. Note that only the component of the particle flux in the direction of $q_i$ changes the norm since the contributions of the perpendicular components correspond to equidensity lines and cancel. The number of particles in an infinitely long rectangular column with unit basis area, oriented toward $q$, is $N = \rho(0,0)/|q_i|$, and it changes in unit time by $\Delta N = q_i v_f N = -v_{f\parallel}\rho(0,0)$ where $v_{f\parallel} = |q_i|d_0/2mv^2$ stands for the length of the component of $v_f$ in the direction of $q_i$. Thus, the spread of inhomogeneities, usually interpreted as the result of diffusive forces, arises actually from decoherence weakened by diffusion.

The time dependence of a generic state, starting with a regular, real $q$ dependence around $q = 0$,

$$\rho_i(x, x_d) = \int \frac{dq}{2\pi} e^{iqx}\chi_{i,q}(x_d),$$ (14)

evolves into:

$$\rho(t, x, x_d) = e^{-\frac{d}{2}(1-e^{-2vt})x_d^2} \int \frac{dq}{2\pi}\chi_{i,q}\left(x_d e^{-vt} + i\frac{\hbar q}{mv}(1-e^{-vt})\right)e^{a_q + iqx + ik_q x_d}.$$ (15)

One finds after a long time evolution, $tv \gg 1$,

$$\rho(t, x, x_d) = e^{ik_f x_d - \frac{d}{2}x_d^2} \int \frac{dq}{2\pi}\chi_{i,q}\left(i\frac{\hbar q}{mv}\right)e^{-t\left(i\frac{fq}{mv} + \frac{d_0\hbar q^2}{2m^2v^2}\right)}$$ (16)

where the decoherence suppresses the elementary components with $q \neq 0$, yielding:

$$\rho(t, x, x_d) \approx \frac{mv}{\sqrt{2\pi d_0 \hbar t}}\chi_{i,0}(0)e^{-t\frac{f^2}{2d_0\hbar} + ik_f x_d - \frac{d}{2}x_d^2}.$$ (17)

The exponential environment-induced suppression of a fixed $q$ is weakened into a prefactor $td_0/mv^2$ by the small $q$ contributions, and the drift due to the external force is generated, together the Gaussian decoherence. The impact of the external force is enhanced

by the small $q$ contributions, and the emerging singularity as the decoherence is turned off is expected due to the instabilities such a force generates in closed dynamics.

It is instructive to look into the norm of the state for $\nu t \gg 1$,

$$
\int dx \rho(t,x,0) = \int \frac{dq}{2\pi} \chi_{i,q}\left(i\frac{\hbar q}{m\nu}\right) e^{i\frac{fq}{m\nu^2} + \frac{q^2}{2}\left[\frac{\hbar}{2m^2\nu^3}(3d_0 - d_2\nu^2) - \frac{\hbar\xi}{m\nu^2}\right] - t(i\frac{fq}{m\nu} + \frac{d_0\hbar q^2}{2m^2\nu^2}) + iqx}.
\tag{18}
$$

A spread of $\Delta q$ of $\chi_{i,q}(x_d)$ around $q = 0$ makes the norm suppressed for $t \gg m^2\nu^2/d_0\hbar\Delta q^2$. The only possibility to preserve the norm is to have a singular peak at $q = 0$, $\chi_{i,q}(i\hbar q/m\nu) \sim \delta(q)$. In other words, elementary components with any inhomogeneity are depopulated by the combined effect of decoherence and dissipation.

## 6. Summary

The symmetry properties of the open dynamics were discussed in the present work. It was pointed out that the mixed states of a closed system enjoy a reduplicated symmetry, $G \to G \otimes G$. Such an enlarged symmetry is reduced back to the original symmetry group or to one of its subgroups when the interaction with the environment is taken into account. The elementary building blocks of the pure states of a closed system are the irreducible representation of the symmetry group. This concept can be extended over mixed states by considering the irreducible representation of the reduplicated symmetry group in the Liouville space of operators. The reduplication of the symmetry group offers a simple way to distinguish open and closed dynamics: the symmetry is restricted to the diagonal subgroup for open dynamics, $G \otimes G \to G$.

The elementary building blocks of an open system can be split into two classes. The first class contains operators with a nonvanishing trace. At least one first class operator must be in the density matrix. A mixed state may contain second class elementary components with a vanishing trace. The time dependence of this latter component is not restricted by the conservation of the total probability.

The importance of the elementary blocks in the dynamics is that the irreducible representation spaces remain closed during the time evolution, offering an important simplification in solving the equation of motion for the density matrix [14]. However, the construction of the irreducible representations using the more complete Galilean or Lorentz groups remains a challenging problem owing to the noncommutativity of the symmetry groups. For instance, in the case of rotational symmetry, one faces the problem of finding all irreducible tensor operators and their matrix elements in the space of pure states.

The irreducible representations of the symmetries were discussed in the context of nonrelativistic quantum mechanics, and it is natural to raise the question about many-body systems, handled by quantum field theory. There seems to be no conceptual difficulty to extend the discussion to the Fock space of free quantum field theories after the representations have been found in first quantized quantum mechanics. However, interactions may generate unexpected representations and open up an interesting, difficult problem.

**Author Contributions:** Conceptualization, J.P. and I.R.; writing—review and editing, J.P. and I.R. All authors have read and agreed to the published version of the manuscript.

**Funding:** This research received no external funding.

**Conflicts of Interest:** The authors declare no conflict of interest.

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
