# Peer review of "Elementary Open Quantum States"

_symmetry, doi:10.3390/sym13091624_

Round 1

Reviewer 1 Report

In this paper, the authors discuss symmetry properties of the open dynamics. The role of symmetries in separating closed dynamics from open dynamics is presented. It is shown that the mixed states of a closed dynamics supports a reduplicated symmetry, which is reduced back to the subgroup of the original symmetry group when the dynamics is open.

As for the results evaluation, the weakest point of this paper, in my opinion, is potentially low significance. I was not able to see how the presented symmetry study would lead to further advanced in open and closed system dynamics studies. Therefore there might be a lack of interest from the broad reader audience.

However, importantly, I believe this paper is still valuable for publication as the mathematical studies that are performed are convincing. These studies might be useful for future work.

As for the rest, the paper is well-written, I do not have any corrections to suggest.

Author Response

We thank for the Referee's attention in reading the paper.

Reviewer 2 Report

The paper describes the formulation of quantum dynamics based on the master equation for open quantum systems. The authors notice that the mixed states of a closed dynamics supports a doubled symmetry, which is reduced to the original symmetry subgroup when the dynamics is open. Also, they show that the elementary building blocks of an open system can be split into two classes, which simplifies to solve the equation of motion. This feature is demonstrated through the translational-invariant harmonic dynamics in an open system.

I think that this paper is well written based on the rigid mathematical analysis and provides an important step toward treating complicated quantum dynamics in open systems. I thus would like to recommend the paper for publication. I just mention one remark about the applicability for more realistic problems. It is helpful for readers if the authors would give some comments what hurdles prevent the usage of the present formulation in e.g, the final section. 

Author Response

We thank for the Referee's attention in reading the paper and the useful remark. 

The complications arise from the non-commutativity of the symmetry group, a remark inserted in the lines 262-267.

The changes are set in red.

Reviewer 3 Report

My comments and suggestions are in the attached file report.pdf

Author Response

We thank for the Referee's attention in reading the paper and the useful remarks. 

- G\otimes G: The notation has been explained in the introduction at lines 50-53.

- QFT: This is a very interesting question. I believe that free field theories pose no problem however interaction may produce new representations. This point is mentioned in the last paragraph, starting with line 268.

- Weighted average of x: The Referee is completely right, one may consider a weighted average of x_+ and x_- for x. But it would amount only to a redefinition of the x_d-dependence of D in (7). The advantage of this choice is that x has well defined time inversion parity which in turn simplifies the coordinate Green functions.

The changes are set in red.